# Adaptive Modular Convolutional Neural Network for Image Recognition

**DOI:** 10.3390/s22155488

**Published:** 2022-07-22

**Authors:** Wenbo Wu, Yun Pan

**Affiliations:** School of Computer and Cyberspace Security, Communication University of China, Beijing 100024, China; cuc_wwb2022@163.com

**Keywords:** image recognition, feature extraction, modular, gate unit

## Abstract

Image recognition has long been one of the research hotspots in computer vision tasks. The development of deep learning is rapid in recent years, and convolutional neural networks usually need to be designed with fixed resources. If sufficient resources are available, the model can be scaled up to achieve higher accuracy, for example, VggNet, ResNet, GoogLeNet, etc. Although the accuracy of large-scale models has been improved, the following problems will occur with the expansion of model scale: (1) There may be over-fitting; (2) increasing model parameters; (3) slow model convergence. This paper proposes a design method for a modular convolutional neural network model which solves the problem of over-fitting and large model parameters by connecting multiple modules in parallel. Moreover, each module contains several submodules (three submodules in this paper) and fuses the features extracted from the submodules. The model convergence can be accelerated by using the fused features (the fused features contain more image information). In this study, we add a gate unit based on the attention mechanism to the model, which aims to optimize the structure of the model (select the optimal number of modules), allowing the model to select an optimum network structure by learning and dynamically reducing FLOPs (floating-point operations per second) of the model. Compared to VggNet, ResNet, and GoogLeNet, the structure of the model proposed in this paper is simple and the parameters are small. The proposed model achieves good results in the Kaggle datasets Cats-vs.-Dogs (99.3%), 10-Monkey Species (99.26%), and Birds-400 (99.13%).

## 1. Introduction

The research of image recognition has shifted from traditional artificial feature extraction to feature extraction based on neural networks, and the process of neural network learning features from a large amount of data does not need too much manual participation. With the introduction of deep learning [1], convolutional neural networks (CNN) are widely used in computer vision recognition tasks because CNN has powerful feature extraction ability and can extract senior semantic information from images. Therefore, the goal has shifted to designing a simple, efficient and high-accuracy model.

Increasing the scale of the model is widely used to obtain better recognition accuracy, and the most common method is to increase the depth or width of the network. For example, VggNet [2] and ResNet [3] can be extended in network depth by using more layers; WRN [4] can be extended in network depth width by using more layers, and recently, EfficientNet [5] is extended from Efficientnet-B0 to Efficientnet-b7 by model scaling. Although the models designed above obtain good recognition accuracy, the parameters of the models increase, and the models’ ability of fitting improves, which may lead to over-fitting. At the same time, the increase in network depth will also generate problems such as gradient exploding/vanishing. Finally, the result will be sub-optimal identification accuracy and efficiency.

In this paper, we study and think about how to design a structure of a model to achieve higher accuracy and efficiency. It is observed that the previous structure of the model is a single module. Can a structure of a model with a multi-module be designed to achieve better accuracy and efficiency? In our study, we find that it is important to improve the feature extraction ability of the model while reducing the complexity of the model. On this basis, we design a simple and efficient structure for the model. Different from previous works, our network structure does not deepen the depth of the network. In the model, multiple convolution blocks are stacked together in parallel, which can enhance the learning ability of the network without expanding the network depth. Due to the simple design of each convolution block, the number of parameters of the model can be greatly reduced. The FLOPs increase as the number of modules in the model increases. However, we find that not all modules contain a large number of features that are beneficial for image recognition. Therefore, we use a gate unit based on the attention mechanism to select modules that are beneficial for recognition and determine an optimum structure of the model. This approach can also dynamically reduce the FLOPs of the model.

We make two contributions. First, we propose a modular convolutional neural network that connects multiple simple convolutional modules in parallel, each of which contains multiple submodules, each of which extracts different image features. Second, we use a gate unit based on the attention mechanism to dynamically reduce the FLOPs of the model and optimize the parameters of the model (the number of modules) to select an optimum structure for the model finally.

## 2. Related Work

### 2.1. Image Recognition

Image recognition methods are divided into traditional methods and deep learning methods. Early work mainly focused on traditional methods, which generally include two stages: image feature extraction and classifier classification. (1) Image feature extraction: it is usually necessary to extract a large number of local features from an image and use these features for image recognition. Refs. [6,7,8,9,10,11] are commonly used local feature extraction algorithms; (2) classifier classification: the image after feature extraction can be represented by a vector of fixed dimension, and the next step is to recognize the image by a classifier. Refs. [12,13,14] are commonly used classifier algorithms. Traditional image recognition methods extract shallow features of the image and cannot contain senior semantic information present in the image. Also, each method is for a specific application and has poor generalization ability and robustness.

With the proposal of deep learning [1], deep convolutional neural networks are highly superior in feature representation, which integrates image feature extraction and classifier classification in an end-to-end manner, and have achieved great success in image and video recognition. AlexNet network proposed by Krizhevsky et al. [15] won the 2012 ImageNet Large Scale Visual Recognition Challenge (ILSVRC) winner of the year. It was also the first application of the convolutional neural network to image classification. VGG nets were proposed by Simonyan et al. achieved second place in ILSVRC 2014 [2]. VGG net improves network performance by increasing the number of network layers. Its framework is now widely used in transfer learning and other network structures that require training due to its simple structure and ease of understanding. Szegedy et al. who proposed GoogleNet won the 2014 ILSVRC [16]. This network introduced the Inception structure, which broke the fixed form of each computing unit of traditional deep neural networks. the structure also shows good results in the follow-up work [17,18,19]. As the network deepens, deep networks suffer from gradient vanishing or exploding, making the model difficult to train. He et al. proposed ResNet network [3] and achieved five firsts at ILSVRC 2015 and COCO 2015. This network proposes residual learning, which effectively solves the problem of gradient vanishing. The emergence of residual learning ensures the stability of network training while deepening the network depth to improve the performance of the model. Similarly, Tan et al. proposed a new model scaling method that uses a simple and efficient compound scaling coefficient to scale depth, width, and resolution uniformly. The series of models obtained after scaling is called EfficientNet [5], which is more efficient and faster than the previous models in terms of accuracy.

### 2.2. Modularity Idea

Modularity is an intuitive, biologically inspired concept. In biological brains, specific information is often processed in fixed regions to accomplish specific functions. In recent years, with breakthroughs in deep learning, the modularity idea has been applied to various fields. Zhang et al. proposed a deep hierarchical multi-patch network based on spatial pyramid matching [20]. Instead of using traditional downsampling, this method uses sectional image segmentation instead, so that the input image of each branch has the same resolution and achieves image deblurring. Dosovitskiy et al. were inspired by Transformer in natural language processing (NLP) and applied Transformer directly to images [21]. This method splits the image into blocks as Transformer’s input and trains the image classification network in the form of supervised learning. He et al. proposed a scalable self-supervised learning method Masked Autoencoders (MAE) [22], which splits the image into blocks, and randomly selects a few subblocks of the image as network input (the rest of the subblocks are masked), and then reconstructs the missing pixels. Carion et al. proposed an issue that treats the object detection task as an image-to-set [23]. After the input image is subjected to CNN feature extraction, the extracted feature blocks are used as input to Transformer, which finally predicts an unordered set containing all objects. Zheng et al. first used the Transformer model for semantic segmentation, which is similar to the method in [21], and finally obtained segmentation prediction results [24].

Tan et al. proposed a weighted bi-directional feature pyramid network (BiFPN) module [25], which can improve the accuracy of object detection by stacking BiFPN modules in series. Newell et al. propose an Hourglass Module [26]. The entire network structure is stacked with many hourglass modules, which enables the network to repeat the bottom-up and top-down processes continuously.

Some of the above methods are to divide the whole image into blocks and send them to the network for processing. The other parts are to stack the convolutional modules horizontally to improve accuracy. Inspired by the above methods, this paper proposes a modular convolutional neural network. The design method of the model is to stack shallow networks longitudinally, and the details will be elaborated in Section 3.1.

### 2.3. Attention Mechanism

In the case of limited computing power, the conscious decision to allocate computing resources to the more important task is to improve system performance. The attention mechanism comes from a 2014 Google DeepMind article [27], it popularized the attention mechanism by developing a new attention-based task-driven neural network vision processing framework. Inspired by the attention mechanism, Bahdanau et al. first applied attention mechanism to the field of NLP in their paper [28]. In the 2017 Google Machine Translation team publication [29], network structures such as recurrent neural network (RNN) and CNN were completely discarded, and only attention mechanisms were used for machine translation tasks. As the attention mechanism can globally capture connections, it is also widely used in computer vision tasks. In 2018, Du et al. put forth a model that introduces a local feature interaction-aware self-attentive mechanism by using the idea of principal component analysis (PCA) for reference [30] and embedded the model into the CNN network. Hu et al. combined the attention mechanism with the convolutional neural network and won the ImageNet2017 competition [31]. In 2019, Wang et al. put forward their attention module [32] to assign pixel-level aggregation weights on each adjacent frame for video deblurring. In 2020, Zhong et al. proposed a global spatio-temporal attention module to fuse the effective hierarchical features of past frames and future frames [33] to help better deblur current frames. Inspired by the attention mechanism, this paper proposes a gate unit based on the attention mechanism to optimize the structure of the model, the details of which will be elaborated in Section 3.2.

## 3. Architectural Details

In this section, we first detail the structure of the model of the proposed modular convolutional neural network, and then introduce the structure of the gate unit and show how to optimize the structure of model.

### 3.1. Modular Neural Network Architectures

#### 3.1.1. The General Structure of the Model

This paper proposes a design method of the modular convolutional neural network, whose idea is similar to ensemble learning. The overall structure of the model is shown in Figure 1. The model consists of M modules, which are stacked in parallel. Each module is divided into n submodules, and 3 submodules are used in this paper because the preprocessing process of each image falls into three parts. Details will be described in Section 3.1.2. Since each main module gathers the features of all the submodules it contains, and each main module contains different features that are helpful for classification, the features contained in all the main modules are aggregated again to obtain the final classification features. However, not every module contains many features conducive to classification. Thus, we design a gate unit to optimize the model structure, and its purpose is to reduce the model parameters and determine the number of modules.

The proposed model can improve the learning ability of the model without increasing the depth of the network, and the parameters and FLOPs of the model can be controlled by adjusting the number of modules. For convenience, from now on we represent the model in the form of M${}_{i}$C${}_{j}$ (e.g., M${}_{2}$C${}_{5}$ represents a model with 2 major modules and 5 convolution layers per submodule). Table 1 shows the parameters and FLOPs of our model and some of the models, and it can be seen that the parameters of ResNet-50 are 26 M and the FLOPs are 4.1B. The params of M${}_{3}$C${}_{6}$ are 3.4 M and the FLOPs are 17.6B, with 57 layers in its model. Although the FLOPs of the model increase, the params decrease significantly, which can also greatly reduce the possibility of over-fitting problems.

#### 3.1.2. Introduction to Function Module

In previous single-module models, most models preprocess the data by using methods such as rotating, flipping, and adjusting brightness uniformly to the input image and inputting the processed image to the model for training (as shown in Figure 2a). Such an approach has some problems, which may lead to the loss of some information in the processed image. In this paper, we apply the data preprocessing method separately to the input images (as shown in Figure 2b–d). As there are three images after processing, we use three submodules to extract the features of these three images(as shown in Figure 3). Although the structure of each submodule is the same, the image features extracted by each submodule are different. Submodule 1 extracts the features of the original image; submodule 2 extracts the features of the image after the horizontal flip, and submodule 3 extracts the features of the image after enhancing brightness. Finally, all the features extracted by the submodules are fused. The advantage is that the features are extracted separately so that as much semantic information as possible can be retained for subsequent recognition.

### 3.2. Gate Unit Based on Attention Mechanism

The structure of the Gate Unit is shown in Figure 4, which aims to: (1) fuse useful information from different modules that are beneficial for classification and ignore useless information to help better classification; (2) optimize the number of modules and eventually determine the optimum model by learning, and (3) dynamically reduce the FLOPs of the model. Inspired by the Squeeze-and-Excitation (SE) block [31], a gate unit based on the attention mechanism is proposed. The input of this structure are M senior features, and then M weights are learned by attention module, and finally the senior features are weighted and fused to obtain the final global features as follows: (1)$$W_{i}=A\left(f_{i}\right)$$
(2)$$Output=W_{1}\times f_{1}+W_{2}\times f_{2}+\cdots +W_{M}\times f_{M}$$
wherein *i* value is 1, 2, …, M; function *A* is the attention module; and *f${}_{i}$* is the senior feature of the $i^{\mathrm{th}}$ module. We set a threshold α. If *W${}_{i}$* is smaller than this threshold, the information contained in the $i^{\mathrm{th}}$ module has negligible impact on improving the accuracy of recognition, and then the module can be deleted for the purpose of optimizing the structure of model.

## 4. Experimental

### 4.1. Datasets

**Cats-vs.-Dogs.** The dataset is derived from the kaggle big data competition and is a classic image recognition task where the images in the dataset are derived from the real world. The dataset comprises a training set and a test set. The training data contains 12,500 images of cats and 12,500 images of dogs, and the test data contains 12,500 images of cats and dogs.

**10-Monkey Species.** The dataset is derived from the kaggle dataset, which is intended to be used as a test case for fine-grained classification tasks. The dataset comprises a training set with 1370 images and a validation set with 272 images.

**Birds-400.** The dataset is derived from the kaggle dataset, which is a very high-quality dataset with only one bird in each image, which usually occupies at least 50% of the pixels in the image. The dataset comprises a training set, a validation set, and a test set. The training set contains 58,388 images, the validation set contains 2000 images, and the test set contains 2000 images.

### 4.2. Training

All the networks are trained using a method for stochastic optimization (Adam) optimizer [39] except for the Vgg16 network, which uses the Stochastic Gradient Descent (SGD) optimizer. We conduct experiments in NVIDIA GeForce RTX 3090, and the batch size is set as 24 to train 50 epochs, 90 epochs, and 90 epochs, respectively, in Cats-vs.-Dogs, 10-Monkey Species, and Birds-400 datasets. The initial learning rate is set to 0.0001. In the network, we add a dropout layer after every three convolutional layers for regularization. The parameter is set to 0.25, and the initial values of the number of modules are set to M = 3 in Cats-vs.-Dogs and 10-Monkey Species datasets; the initial values of the number of modules is set to M = 5 in the Birds-400 dataset. The threshold of the gate unit is set to. When the parameter *W${}_{i}$* of the gate unit is less than this threshold, we consider the module contains less information that can help in classification and the module can be removed. We resize the input images in the network to 224 × 224 by using bilinear interpolation and use the data enhancement method of horizontal flip and brightness enhancement for the input images. Finally, the model is evaluated in three datasets (as shown in Table 2).

### 4.3. Experimental Result

#### 4.3.1. Classification Results

Table 3 and Table 4 and Figure 5 show the test results in the Cats-vs.-Dogs and 10-Monkey Species datasets. We use the M${}_{3}$C${}_{13}$ model with a gate unit to train the network, which is denoted as M${}_{3}$C${}_{13}$-reg for convenience. Compared to lightweight networks, such as MobileNetV2, although the parameters of the model are higher than that of MobileNetV2; it achieves higher accuracy. Compared to some classical models, such as Vgg16 and ResNet50, the parameters of the model are significantly reduced and higher accuracy is achieved. When we train in the Cats-vs-Dogs and 10-Monkey Species datasets, the vectors of gate units learned by the network were all (1, 1, 1), wherein 1 refers to on and 0 refers to off. It shows that each module of the model contains more information that helps in classification. We also test in the M${}_{2}$C${}_{13}$ model and finds that its accuracy is lower than that of the M${}_{3}$C${}_{13}$-reg model. Therefore, we consider the model as the optimum model when the number of modules is 3 (M = 3) in the Cats-vs.-Dogs and 10-Monkey Species datasets.

Table 5 and Figure 5 show the test results in the Birds-400 dataset. As of a large number of classes in the dataset, we choose M${}_{5}$C${}_{13}$-reg model to train the network. When we train in the Birds-400 dataset, the vector of the gate unit learned by the network is obtained as (1, 1, 1, 1, 0). This means that there are 4 modules in the model that contain more information that helps to classify and 1 module that contains less information that helps to classify. We also test in M${}_{4}$C${}_{13}$ and M${}_{3}$C${}_{13}$ models and founds that the highest accuracy of recognition is achieved when the model is M${}_{4}$C${}_{13}$. Therefore, we consider that the optimum model is when the number of modules is 4 in the Birds-400 dataset (M = 4).

#### 4.3.2. Experimental Conclusion

Figure 6 shows the variation in the accuracy of each model. We find that the accuracy of the proposed model is low in the first 20 epochs of training. However, when epoch >20, the accuracy of the proposed model increases fast, while the accuracy of the other models increases low. In particular, in the 10-Monkey Species dataset, Table 6 shows the accuracy of the other three models when training 300 epochs, and it can be found that the accuracy of the other three models improves greatly compared to epoch = 90, but the improvement is very slow. However, the proposed model can achieve 99.26% accuracy when training 90 epochs. Therefore, the proposed model can achieve a good recognition quickly during the network training and accelerate the convergence of network.

The proposed model combines the advantages of a lightweight model and a large-scale model. The model has fewer parameters and can produce accurate fit and good generalization. In addition, according to different datasets, the model can learn the optimal structure suitable for the dataset.

## 5. Conclusions

With the development of artificial intelligence, the application of image recognition is more and more extensive, for example, conducting geological surveys and disaster prediction through aerial remote sensing image recognition, helping public security departments solve crimes by identifying crime scenes photos, fingerprints and portraits. However, the existing models either have good precision but high model complexity or have low model complexity but low precision. Therefore, an efficient and high-precision model is needed. In this paper, we propose a model for modular convolutional neural networks. Unlike most models, this model stacks several modules in a parallel manner without increasing the depth of the network and determines the optimum model structure by learning based on a large number of experiments and gate units.

The advantages of the model are (1) it can significantly reduce the parameters of the model and better avoid over-fitting problems; (2) it uses a multitasking approach to improve the learning ability of the network; (3) it uses a gate unit based on attention mechanism to optimize the model structure by learning and also to reduce the FLOPs of the model dynamically. However, the model also has some shortcomings: (1) the FLOPs of the model increase as the number of modules increases, which requires more GPU with higher performance to handle; (2) the current experiments are classified into 2, 10, and 400 categories, and further validation and improvement on datasets with more categories are yet to be performed; (3) the number of modules M and the threshold of the gate unit are set based on experimental and empirical and still need to be tested on a large scale to discover certain setting rules. Therefore, we hope to find better optimization methods to optimize the model in the next research and to find suitable tasks for the model in other computer vision fields.

## Figures and Tables

**Figure 1 sensors-22-05488-f001:**
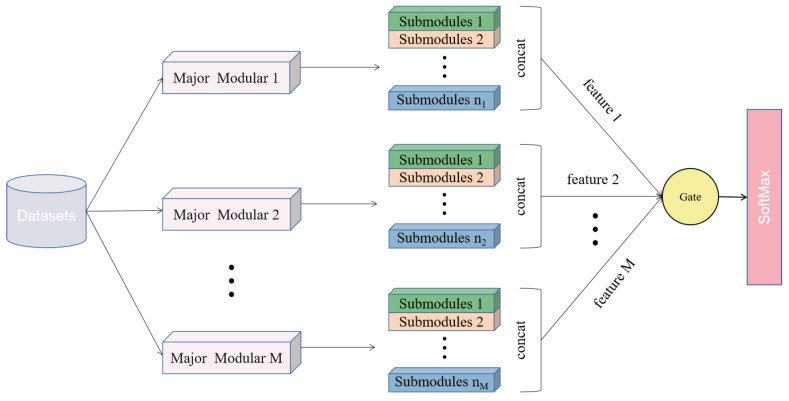
Overall Structure of Model. M refers to the number of modules. n${}_{M}$ refers to module M containing n submodules.

**Figure 2 sensors-22-05488-f002:**
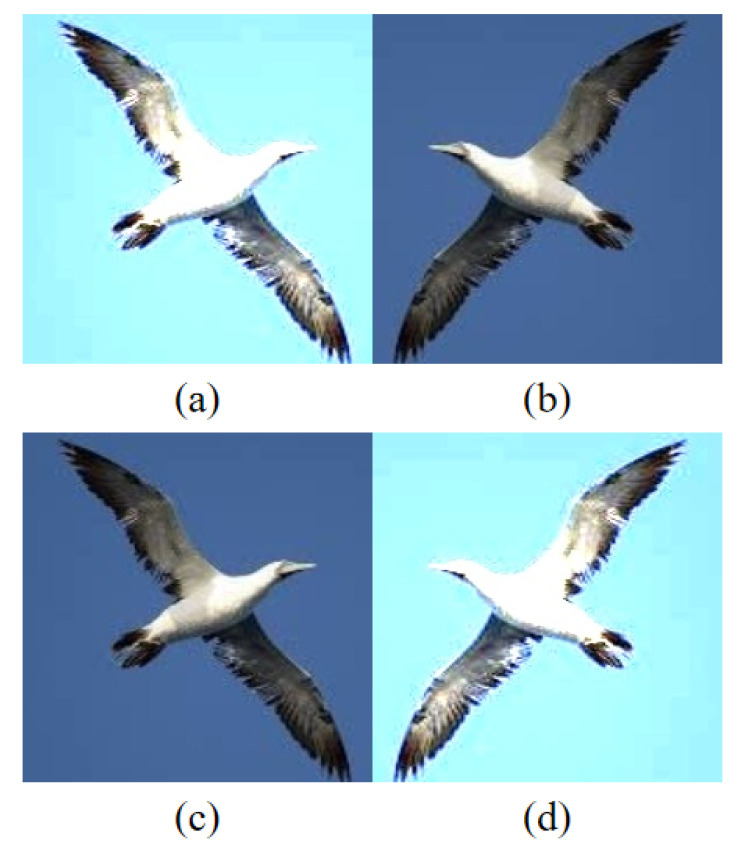
Data Preprocessing. (**a**) Enhancing brightness after flipping the original image. (**b**) Original image. (**c**) Flipped image. (**d**) Image with enhanced brightness.

**Figure 3 sensors-22-05488-f003:**
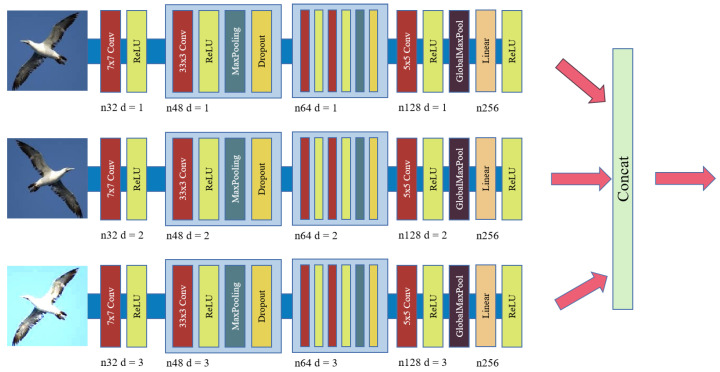
Diagram of Submodules.

**Figure 4 sensors-22-05488-f004:**
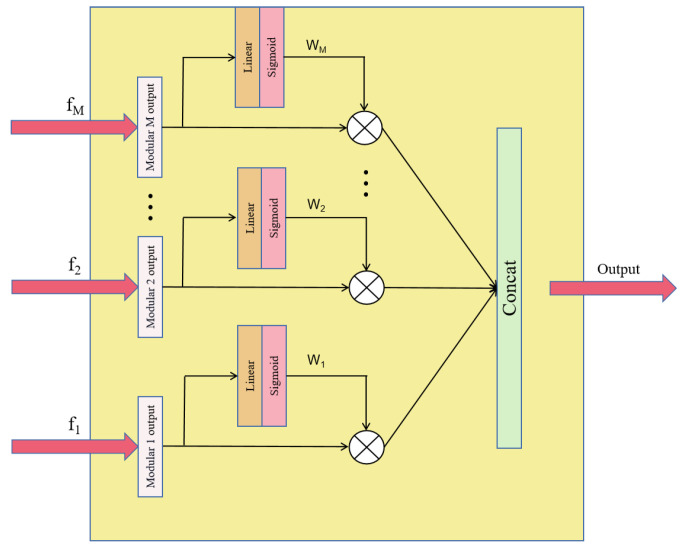
Feature Weighted Fusion Based on Attention Mechanism.

**Figure 5 sensors-22-05488-f005:**
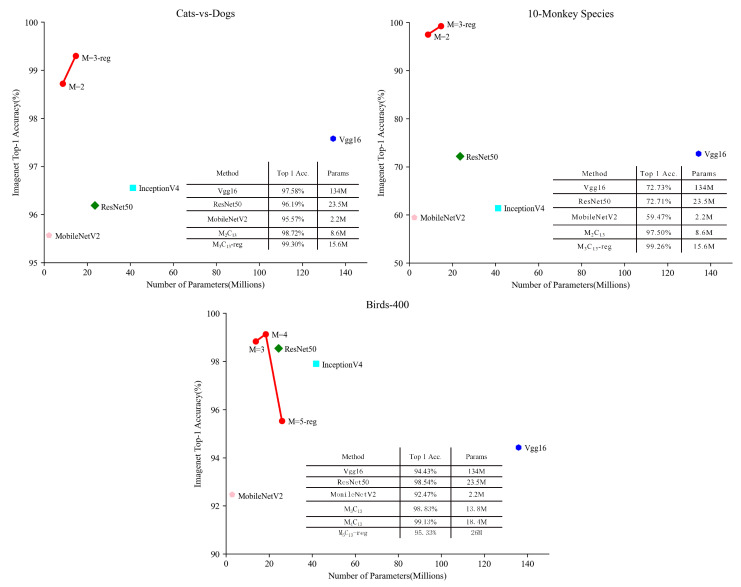
Model Parameters vs. Datasets Accuracy.

**Figure 6 sensors-22-05488-f006:**
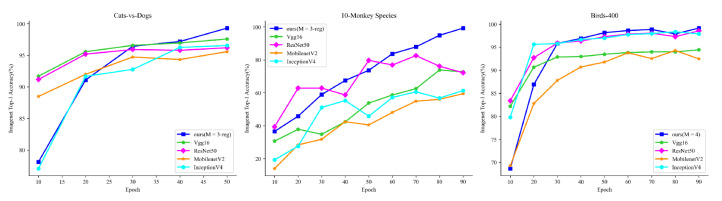
The Accuracy of Each Model.

**Table 1 sensors-22-05488-t001:** Parameters and FLOPs of Model.

Method	Params	FLOPs
ResNet-50 [3]	26 M	4.1B
PolyNet [34]	92 M	35B
Inception-v4 [19]	48 M	13B
ResNeXt-101 [35]	84 M	32B
PNASNet [36]	86 M	23B
NASNet-A [37]	89 M	24B
AmoebaNet-A [38]	87 M	23B
M${}_{3}$C${}_{6}$	3.4 M	17.6B
M${}_{5}$C${}_{6}$	5.7 M	29.3B

**Table 2 sensors-22-05488-t002:** Datasets.

Dataset	Training Size	Validation Size	Test Size	Classes
Cats-vs.-Dogs	12,500	-	12,200	2
10-Monkey Species	1370	272	-	10
Birds-400	58,388	2000	2000	400

**Table 3 sensors-22-05488-t003:** Evaluation Results in the Cats-vs.-Dogs Dataset.

Method	Epoch	Batchsize	Optimization	Params	FLOPs	Accuracy
Vgg16 [2]	50	24	SGD	134.27 M	15.48B	97.58%
ResNet50 [3]	50	24	Adam	23.51 M	4.11B	96.19%
MobileNetV2 [40]	50	24	Adam	2.23 M	0.31B	95.57%
InceptionV4 [19]	50	24	SGD	41.15 M	12.29B	96.55%
M${}_{2}$C${}_{13}$	50	24	Adam	8.59 M	34.22B	98.72%
M${}_{3}$C${}_{13}$-reg	50	24	Adam	14.66 M	51.33B	99.30%

**Table 4 sensors-22-05488-t004:** Evaluation Results in the 10-Monkey Species Dataset.

Method	Epoch	Batchsize	Optimization	Params	FLOPs	Accuracy
Vgg16 [2]	90	24	SGD	134.30 M	15.48B	72.73%
ResNet50 [3]	90	24	Adam	23.53 M	4.11B	72.21%
MobileNetV2 [40]	90	24	Adam	2.24 M	0.31B	59.47%
InceptionV4 [19]	90	24	SGD	41.16 M	12.29B	61.36%
M${}_{2}$C${}_{13}$	90	24	Adam	8.61 M	34.22B	97.50%
M${}_{3}$C${}_{13}$-reg	90	24	Adam	14.68 M	51.33B	99.26%

**Table 5 sensors-22-05488-t005:** Evaluation Results in the Birds-400 Dataset.

Method	Epoch	Batchsize	Optimization	Params	FLOPs	Accuracy
Vgg16 [2]	90	24	SGD	135.90 M	15.48B	94.43%
ResNet50 [3]	90	24	Adam	24.33 M	4.11B	98.54%
MobileNetV2 [40]	90	24	Adam	2.74 M	0.31B	92.47%
InceptionV4 [19]	90	24	SGD	41.76 M	12.29B	97.89%
M${}_{3}$C${}_{13}$	90	24	Adam	13.81 M	51.33B	98.72%
M${}_{4}$C${}_{13}$	90	24	Adam	18.41 M	68.44B	99.13%
M${}_{5}$C${}_{13}$-reg	90	24	Adam	25.97 M	85.55B	95.33%

**Table 6 sensors-22-05488-t006:** Accuracy of the other Three Models when Training 300 Epochs.

Method	Epoch	Accuracy
Vgg16 [2]	300	88.26%
ResNet50 [3]	300	90.15%
MobileNetV2 [40]	300	81.82%
InceptionV4 [19]	300	86.74%
M${}_{3}$C${}_{13}$-reg	300	99.26%

## Data Availability

The data is available on the Kaggle website.

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
