# Peer review of "Adaptive Modular Convolutional Neural Network for Image Recognition"

_sensors, 2022, doi:10.3390/s22155488_

Round 1

Reviewer 1 Report

The paper describes a modular approach to the classical problem of image classification by neural networks. The authors propose a special kind of combination of different modules and submodules (conv nets, MaxPooling layers,…) which are then comnbined with a gate unit that uses the well known attention mechanism. The overall combination of modules may be new (at least I have not seen it before), but I can not tell this for sure. The description if the approach is mainly good, however, I would not be able to build up the same model from the description. It would be helpful if the authors would provide their code online so that the read is able to look up every detail. Code sharing is common practice in this area.

Here are a few questions and comments:

- One of the motivations given in the introduction is the problem of overfitting. However, recent research has shown that deep learning models with much more parameters than data produce exact fitting and good generalization at the same time which challenges this motivation (papers can be found under the search term "benign overfitting").

- How is the proposed model different from a shallow conv-net with many channels (and followed by a gate unit as in the paper)?

- I do not understand what is meant by "FLOPs of the model". To my understanding, FLOPs are a measure of computational effort…

- In Figure 2 I do not get what is shown in subfigure (a). How is this "stacking all preprocessing models"?

- The typesetting in lines 180-181 is a bit weird (math terms not set in math style, missing spaces…).

- Why is Vgg16 not trained with adam?

In general, the use of English could be improved (missing articles, not always correct word order…).

The performance of the proposed model seems to be good (up to the question I raised above).

Reviewer 2 Report

In this paper, a modular convolutional neural network using use a gate unit based on attention mechanism. Although, there are many problems are advised to modify in this paper as follows:

1. In section 2.2, modularity Idea is inspired, please figure out the idea detail reference come from. As well as the attention mechanism in section  2.3.

2. In section 3, I cannot see the relationship between the module and submodule. The details of the relationship is not descripted.

3. In section 3, multiple modules are parallel connected. How many modules and sub modules are there?

In section 3.2, the value of weights Wi is come from training? Or by parameters setting? Give out the value.

4. In section 4, some parameters setting are not given, such as the number of modules M, the number of submodules n, and the value of weight Wi. In experiment, these key values must be given out to verify the truth of the test.

5. In section 3.1, there are three submodules to get achieve the function of image enhancement. The structures of the three sub modules are identical. Why do you can give out the different task of submodules? Just because the different input images? Please give out the reason or change the structure of three submodules.

6. Multiple modules are parallel connected. What is the difference between it and the Inception net? And give out the advantages of this paper. More importantly, in section4, the paper have not given the test with Inception net. As we know, the Inception net is a board widely used parallel net.

7. The training platform is not given in the test.

8. The efficiency analysis and complexity analysis are not give in section3.

9. The conclusion of test results are not given.  

10. The application field is not given.

Round 2

Reviewer 1 Report

The author's made some changes according to the raised points and I think that the paper has improved. However, I still think that the authors should address two points again:

1. Please explain what is meant by FLOPS here. Is that the number of floating point operations for one pass through the network?

2. Please explain more clearly what "stacking" of networks means and how it's different from using more channels in parallel.

Reviewer 2 Report

Revision completed as expected. Accept.
